# Marek’s Disease Virus Infection of Natural Killer Cells

**DOI:** 10.3390/microorganisms7120588

**Published:** 2019-11-20

**Authors:** Luca D. Bertzbach, Daphne A. van Haarlem, Sonja Härtle, Benedikt B. Kaufer, Christine A. Jansen

**Affiliations:** 1Institute of Virology, Freie Universität Berlin, 14163 Berlin, Germany; luca.bertzbach@fu-berlin.de; 2Department of Infectious Diseases and Immunology, Faculty of Veterinary Medicine, Utrecht University, 3584 Utrecht, The Netherlands; d.a.vanhaarlem@uu.nl; 3Department of Veterinary Sciences, Ludwig-Maximilians-Universität München, 80539 Munich, Germany; sonja.haertle@lmu.de

**Keywords:** NK cells, chickens, cell tropism, CD107, interferon gamma (IFNγ), RB-1B, CVI988, *meq*

## Abstract

Natural killer (NK) cells are key players in the innate immune response. They kill virus-infected cells and are crucial for the induction of adaptive immune responses. Marek’s disease virus (MDV) is a highly contagious alphaherpesvirus that causes deadly T cell lymphomas in chickens. Host resistance to MDV is associated with differences in NK cell responses; however, the exact role of NK cells in the control of MDV remains unknown. In this study, we assessed if MDV can infect NK cells and alter their activation. Surprisingly, we could demonstrate that primary chicken NK cells are very efficiently infected with very virulent RB-1B MDV and the live-attenuated CVI988 vaccine. Flow cytometry analysis revealed that both RB-1B and CVI988 enhance NK cell degranulation and increase interferon gamma (IFNγ) production in vitro. In addition, we could show that the MDV Eco Q-encoded oncogene (*meq*) contributes to the induction of NK cell activation using *meq* knockout viruses. Taken together, our data revealed for the first time that NK cells are efficiently infectable with MDV and that this oncogenic alphaherpesvirus enhances NK cell degranulation and increased IFNγ production in vitro.

## 1. Introduction

Marek’s disease virus (MDV) is an alphaherpesvirus that causes deadly lymphomas in chickens with a mortality of up to 100% [1]. The development of the lymphoproliferative disease is accompanied by clinical signs including immunosuppression, torticollis, ataxia, and paralysis [1]. In infected animals, MDV productively infects chicken B and T cells [2,3,4] and establishes latency mostly in CD4+ T cells [5]. In addition, it has been recently shown that MDV can also infect certain phagocytes and endothelial cells [6,7,8]. Upon infection of chickens, MDV activates both innate and adaptive immune responses [9]. One of the cell types involved in the early MDV-specific immune response are natural killer (NK) cells [10]. 

NK cells are major components of the innate immune system and belong to group I innate lymphocytes. These cells are known to rapidly kill virus-infected cells [11,12] and play a crucial role in tumor immunosurveillance [13]. Activation of NK cells leads to degranulation of cytotoxic granules containing perforin and granzyme and production of various cytokines including interferon gamma (IFNγ) [14]. IFNγ is involved in activation of the adaptive immune system [15,16] and orchestrates the expression of interferon inducible genes [17]. Interestingly, NK cell responses are thought to be involved in resistance to MDV [18], as NK cell activity is elevated in MDV-resistant chickens when compared to susceptible lines [10,19]. Moreover, vaccination against MDV has been shown to enhance NK cell activity, which is associated with a better vaccine protection [20]. Unfortunately, until now very little was known about the interplay of MDV with NK cells and their response to infection. In a previous study, we identified markers that are expressed on chicken NK cells and developed assays to measure NK cell degranulation and killing [21]. Here we used these tools and recombinant viruses to determine if MDV can infect NK cells and if this affects NK cell degranulation and IFNγ secretion.

We demonstrate that both a very virulent strain and the CVI988 vaccine strain efficiently infect primary chicken NK cells. In these infected cultures, we observed an increased NK cell degranulation and IFNγ production. Moreover, we demonstrate that the increased NK cell degranulation is dependent on the expression of the major MDV oncogene *meq* [22]. Our study provides the first evidence that NK cells are efficiently infected, highlighting a so far unknown tropism of this highly oncogenic herpesvirus.

## 2. Materials and Methods

### 2.1. Cells

Chicken embryo cells (CEC) were isolated from 11-day-old VALO specific-pathogen-free (SPF) embryos (VALO Biomedia; Osterholz-Scharmbeck, Germany) as described previously [23]. CECs were maintained in minimal essential medium (MEM, PAN Biotech; Aidenbach, Germany) supplemented with 1–10% fetal bovine serum (PAN Biotech) and 100 U/ml penicillin and 100 μg/ml streptomycin (AppliChem; Darmstadt, Germany) at 37 °C and 5% CO_2_. 

Splenocytes were isolated from 14-day-old Novogen brown embryos, which we obtained from a commercial hatchery (Verbeek; Zeewolde, The Netherlands). At this developmental stage, a population of cells that resemble mammalian NK cells is abundantly present in the embryonic spleen. These cells lack surface expression of T or B cell-specific antigens and are able to kill a NK-susceptible target cell line [24]. We will refer to this population as NK cells. NK cells were maintained in Iscove Modified Dulbecco Medium (IMDM) supplemented with 2% heat inactivated fetal bovine serum, 8% heat inactivated chicken serum (Biochrom; Berlin, Germany), 100 U/ml penicillin, 100 μg/ml streptomycin and 2 mM glutamax I (Gibco; Carlsbad, CA, USA), which we refer to as “NK medium”. NK cells were isolated as previously described [21]. Briefly, spleens were homogenized using a 70 µM cell strainer (Beckton Dickinson (BD); Franklin Lakes, NJ, USA) to obtain a single cell suspension. Viable cells were then purified by Ficoll–Paque density gradient centrifugation [25], resuspended in NK medium and used directly or stored in liquid nitrogen until further use.

### 2.2. Viruses

MDV reporter viruses expressing the green fluorescent protein (GFP) under the control of the herpes simplex virus 1 (HSV-1) thymidine kinase promotor were generated based on the very virulent RB-1B field strain and the vaccine strain CVI988. GFP was inserted into the bacterial artificial chromosome (BAC) backbone replacing the Escherichia coli gpt gene (Eco-gpt). The RB-1BΔmeq virus was generated by en-passant mutagenesis (Appendix A) as described previously [26,27] using the primers provided in Appendix A. All recombinant GFP reporter viruses were reconstituted by transfection of fresh CEC with purified BAC DNA using CaPO_4_ transfection [28]. The viruses were propagated on CEC for up to six passages, and infected cells were stored in liquid nitrogen.

### 2.3. Growth Kinetics and Plaque Assays 

The replication properties of all recombinant viruses were determined by plaque size assays and quantitative PCR (qPCR)-based multi-step growth kinetics. For plaque size assays, one million CEC were infected with 100 plaque-forming units (PFU) of each virus and MDV plaques were analyzed at 6 days post infection (dpi). Images of at least 50 randomly selected plaques were taken and plaque areas determined using Image J (National Institutes of Health, USA). For multi-step growth kinetics, 6 well plates were infected with 100 PFU of each virus in each well. Every day, one well was harvested, and DNA extracted using the Invisorb® DNA Tissue HTS 96 Kit (Stratec Molecular; Berlin, Germany) according to the manufacturer’s instructions. MDV genome copies were determined by qPCR as we described previously [4] using the primers and probes provided in Appendix A. Growth kinetics and plaque sizes were determined in three independent experiments.

### 2.4. Infection of NK Cells

GFP-expressing RB-1B or CVI988-infected CEC and uninfected CEC (mock) were thawed and seeded into a 24 well cell culture plate with PFUs ranging from 2.5 × 10^4^ to 10^5^. In parallel, primary NK cells were labeled with a PKH26 red fluorescent cell linker kit (Sigma-Aldrich; Zwijndrecht, The Netherlands) according to the manufacturer’s protocol. 2 × 10^5^ cells/ml labeled NK cells were added to the uninfected and infected CEC monolayer. Cultures were incubated at 37 °C and 5% CO_2_ for 4 h (CD107 surface expression analysis), or 6 h (analysis of IFNγ production). Additionally, co-cultures of GFP-labeled viruses and unlabeled NK cells were performed, which were incubated for up to 72 h to assess infectivity of these viruses in NK cells. 

### 2.5. CD107 Assay

The CD107 assay was used as previously described with some minor modifications [21]. Briefly, Golgistop (BD) was added to the cells in a final dilution of 1:1000, together with an allophycocyanin (APC)-conjugated mouse anti-chicken CD107 monoclonal antibody (mouse anti-chicken LEP-100, I (IgG1), clone 5G10. Developmental Studies Hybridoma Bank (DSHB); University of Iowa, IA, USA). To separate wells containing NK cells in the absence of virus, phorbol-12-myristate 13-acetate (PMA)/ionomycin (Sigma-Aldrich) and medium were added as positive and negative control respectively, as previously described [21]. After co-culturing virus-infected CEC and NK cells for 4 h at 37 °C/5% CO_2_, non-adherent cells were harvested for flow cytometry.

### 2.6. IFNγ Assay

To analyze interferon gamma (IFNγ) production by NK cells, Golgiplug (BD) was added to each well of the co-cultures in a final dilution of 1:1000 followed by an incubation of 6 h at 37 °C/5% CO_2_ before non-adherent cells were harvested for flow cytometry.

### 2.7. Flow Cytometry

In experiments with unlabeled NK cells, the non-adherent cells were stained with an APC conjugated mouse-anti-chicken CD45 antibody (LT40, IgM, Southern Biotech; Birmingham, AL, USA) for 20 min at 4 °C. Next, cells were washed and resuspended in a fluorescence activated cell sorting (FACS) buffer made of PBS supplemented with 0.5% bovine serum albumin (BSA) and 0.005% NaN_3_. For CD107 measurements, the non-adherent cells were washed and resuspended in FACS buffer.

To analyze IFNγ by intracellular cytokine staining, the non-adherent cells were washed in FACS buffer and permeabilized for 10 min with 0.5 mL of 1 volume of FACS permeabilizing solution (BD), 1 volume of FACS lysing solution (BD), and 8 volumes of water. Next, cells were washed in FACS buffer and stained with an APC-conjugated mouse-IgG1 anti-chicken IFNγ antibody (mAb80), kindly provided by Dr. Lowenthal [29]. After a 20 min incubation at 4 °C, the cells were washed and resuspended in FACS buffer.

At least 50,000 cells in the lymphocyte gate were acquired using a FACS Canto flow cytometer (BD). Data were analyzed with FlowJo (Tree Star Inc.; Ashland, OR, USA). NK cells were selected as either CD45 positive or PKH26 positive cells. Cells that were double positive for CD45 and GFP or PKH26 and GFP were defined as MDV-infected chicken NK cells.

### 2.8. Statistical Analyses

Non-parametric statistical tests were used when the assumption of normal distributed data were not met. A p-value of <0.05 was considered statistically significant. All statistical analyses were performed using the program GraphPad Prism 7.05 (GraphPad Software, Inc.; La Jolla, CA, USA).

## 3. Results and Discussion

### 3.1. Reporter Viruses Replicate to Comparable Levels

To assess the ability of MDV to infect NK cells, we used GFP expressing reporter viruses. To confirm that the reporter viruses replicate in a comparable manner, we performed plaque size assays and multi-step growth kinetics as described above. Plaque size assays revealed that all reporter viruses efficiently replicate at comparable levels. (Figure 1A, Appendix A). We confirmed these results using qPCR-based multi-step growth kinetics (Figure 1B). These results are in line with the replication of previously generated reporter viruses [30,31].

### 3.2. MDV Can Readily Infect Chicken NK Cells

Next, we assessed the ability of RB-1B and the CVI988 vaccine to infect primary NK cells by co-cultivation with infected CEC. To our surprise, we could very efficiently infect chicken NK cells at levels between 20 and 75 percent depending on the dose of infection after 4 and 24 h post infection (Figure 1C,D). The rate of infection was thereby dependent on the PFU present in the inoculum. Higher infection rates were observed for RB-1B (Figure 1C) when compared to CVI988 (Figure 1D), indicating that a virulent virus infects NK cells more efficiently. 

These infection rates were higher than observed in primary B and T cells [32], in primary endothelial cells [8], and in primary macrophages [7]. In those cells, infection rates ranged from 2–20%. Notably, we optimized the B cell infection system since it was published [32] and readily achieve infection rates of about 50% in primary chicken B cells in vitro.

### 3.3. MDV Enhances NK Cell Degranulation and Increases IFNγ Production

To investigate the effect of MDV on NK cell activation, we performed co-cultures of MDV and primary NK cells and analyzed NK cell activation by measuring CD107 surface expression as a proxy for degranulation and the production of interferon gamma (IFNγ) by flow cytometry. Prior to co-cultivation, NK cells were labeled with the fluorescent dye PKH26 to allow efficient discrimination between NK cells and infected CEC. PKH26 positive cells were selected within the single cell population. It is important to note that CD3 is not expressed in these cells as described previously [21]. Based on forward and side scatter, we selected the lymphocyte population and assessed the CD107 surface expression. Next, we analyzed the GFP expression within the CD107 positive cells by gating on the GFP positive or GFP negative cells (Figure 2).

When NK cells were incubated with 100,000 PFU of the virulent strain RB-1B, CD107 expression increased from 1.56% to 3.51% compared to the mock (*p* > 0.05). A similar trend was observed in a co-culture of NK cells with CVI988 (Figure 3A). Furthermore, we investigated if MDV infections induce IFNγ production by intracellular cytokine staining, since the expression of CD107 is known to correlate with IFNγ production [21]. Again, single cells were gated and within that population, we selected the PKH26 positive NK cells. We could readily detect a population of IFNγ positive cells (Figure 2D) and determine the amount of IFNγ positive lymphocytes (Figure 2J) as well as the GFP expression within the IFNγ positive cells (Figure 2K). Here, the production of IFNγ significantly increased upon co-cultures with RB-1B and CVI988 compared to mock (Figure 3B). Taken together, we could demonstrate that RB-1B and CVI988-infected NK cells show higher levels of surface CD107 and an increased IFNγ production when compared to mock-infected cells (Figure 3A,B). Overall, this suggests that NK cells are activated in these cultures. Interestingly, we observed that only the small population of CD107+ cells was infected with MDV, suggesting that the infection could be a consequence of NK cell activation. The same is true for T cell infections by the virus, as it is known that activation is a crucial step for MDV infection of primary chicken T cells [33].

### 3.4. Lower Rates of Activation in the Absence of the Major MDV Oncogene meq

Finally, to provide a possible mechanistic explanation for the enhanced NK cell activation, we investigated the role of the major MDV oncogene *meq* on CD107 expression and IFNγ production. *meq* has a plethora of functions including up and downregulation of cellular and viral genes and impacts a variety of disease-associated pathways; however, its full functional significance is not yet completely understood, despite being the most studied MDV gene [34,35]. 

To assess the role of *meq* in the NK cell activation, we used a recombinant virus with a deletion of the *meq* gene. We could demonstrate that the CD107 surface expression on chicken NK cells is comparable to the mock-infected cells upon infection with the *meq* deletion mutant, while RB-1B and CVI988 that express *meq* induced CD107 (Figure 3C). Interestingly, the levels of IFNγ production were not affected in the absence of *meq* (Figure 3D). 

Although both degranulation and IFNγ production can be induced upon NK cell activation, these functions are not necessarily executed by one NK cell subset. Different NK cell subsets have been described in humans and mice, based on marker expression and function [15,36]. In chickens however, characterization of NK cells subsets is still in its infancy [37]. Differential effects of MDV wild type and mutant viruses on various NK cell subsets could be a likely explanation for the observed effect of the *meq* deletion mutant. 

Our data suggests that *meq* influences NK cell activation and thereby adds another piece to the puzzle of the wide range of functions of this virulence factor.

## 4. Conclusions

Taken together, we demonstrated that chicken NK cells can be efficiently infected with MDV and thereby shed light on an extended host cell tropism for the virus. Furthermore, our in vitro data demonstrate enhanced NK cell degranulation and release of IFNγ, which could be part of their antiviral response to MDV infections. Future studies will address the role of NK cells in MDV infection in vivo using NK cell knockout chickens that are hopefully available in the years to come [38]. 

## Figures and Tables

**Figure 1 microorganisms-07-00588-f001:**
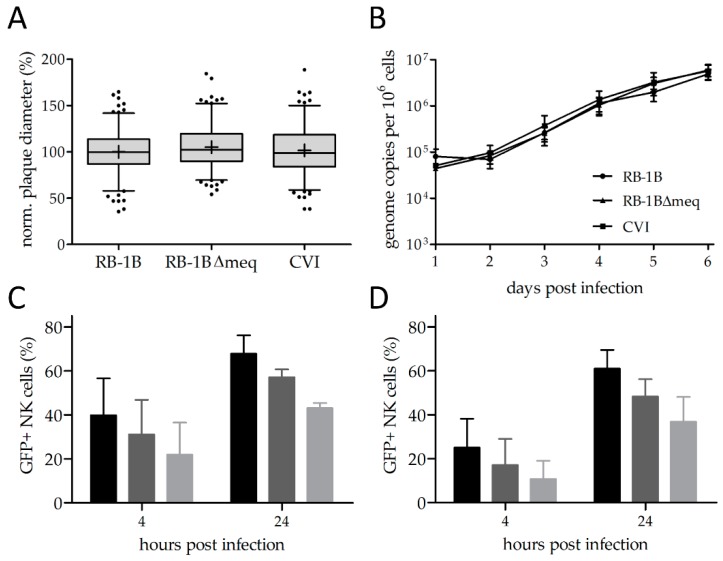
Virus in vitro characterization and natural killer (NK) cell infection rates. (**A**) Plaque size assays and (**B**) multi-step growth kinetics of indicated viruses (*p* > 0.05, one-way analysis of variance (ANOVA)). (**C**) NK-cell infection rates upon infection with the very virulent RB-1B strain and (**D**) the Marek’s disease virus (MDV) vaccine strain CVI988 at 4 and 24 h post infection with different plaque forming units (PFU): black = 10^5^ PFU, dark gray = 5 × 10^4^ PFU, light gray = 2.5 × 10^4^ PFU. Error bars represent the standard error of the mean.

**Figure 2 microorganisms-07-00588-f002:**
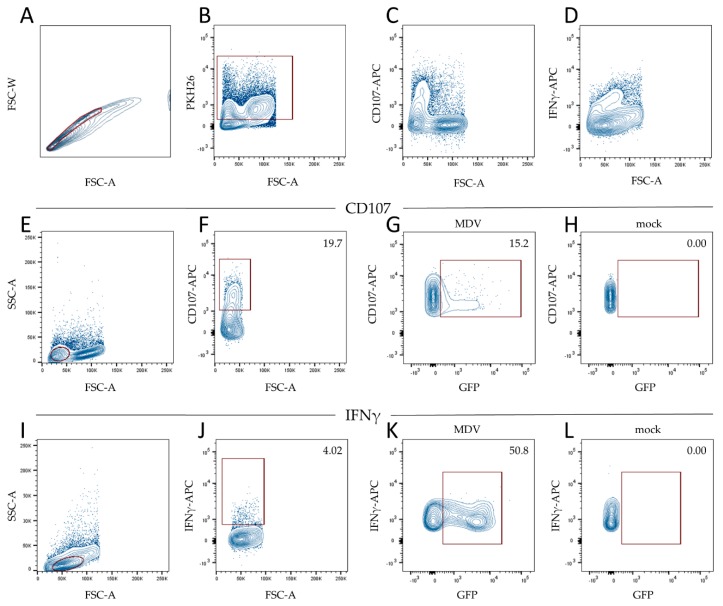
Gating strategy to determine CD107 positive and interferon gamma (IFNγ) positive NK cells. (**A**) We gated single cells based on forward scatter-area (FSC-A) and forward scatter-width (FSC-W). (**B**) Within the single cells, PKH26 labeled NK cells were gated for the subsequent analysis of CD107 expression (D–G) and IFNγ expression (H–K). (**C**) We found that CD107 was expressed on cells with a low forward scatter. (**D**) IFNγ plotted against FSC-A scatter showed one population of cells that expressed IFNγ. (**E**) Lymphocytes were gated based on forward and side scatter. (**F**) Within this gate, we selected the CD107 positive cells and determined GFP positive cells in (**G**) MDV-infected and (**H**) mock-infected cultures. (**I**) Lymphocytes were gated based on FSC-A and side scatter-area (SSC-A). (**J**) We then selected the IFNγ producing cells within the lymphocyte gate and subsequently determined GFP positive cells in (**K**) MDV-infected and (**L**) mock-infected cultures.

**Figure 3 microorganisms-07-00588-f003:**
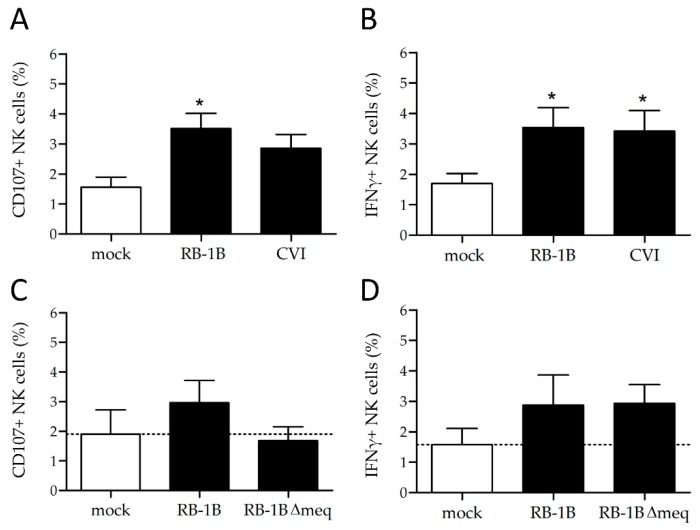
NK cell activation rates and IFNγ production. (**A**) Assessment of CD107+ cells and (**B**) IFNγ production upon infection of NK cells with RB-1B vs CVI988. The same assays were performed to determine differences in (**C**) the percentage of CD107+ cells and (**D**) IFNγ production comparing infections of NK cells with RB-1B and the RB-1BΔmeq mutant. Asterisks indicate statistically significant differences to mock-infected cells (* *p* > 0.05, Kruskal–Wallis test). Error bars represent the standard error of the mean.

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
