# Peer review of "Marek’s Disease Virus Infection of Natural Killer Cells"

_microorganisms, 2019, doi:10.3390/microorganisms7120588_

Round 1

Reviewer 1 Report

The study by Dertzbach et al. investigated if Marek’s disease virus (MDV) could infect natural killer (NK) cells and affect their response to infection. They demonstrated that MDV efficiently infected primary chicken NK cells, leading to interferon gamma production and cell degranulation. By using meq knockout viruses, the authors showed that this gene could be involved in NK activation.

Data are original and interesting and the manuscript is well written.

Comments:

Methods:

2.2. Viruses section: More details, including maps, on the generation of MDV reported viruses should be provided. In addition, the RB-1BΔmeq strain should be described.

Results and discussion:

Page 6, last paragraph. The authors stated that infection rates in NK cells were higher than in primary B and T cells, in primary endothelial cells, and in primary macrophages. However, these data were obtained from studies published in the literature. The authors should perform comparative experiments or, at least, describe more in detail with quantitative data the results reported in the literature.

Author Response

Dear reviewer 1, we would like to thank you for evaluating our manuscript. We have addressed all of your concerns and modified the manuscript accordingly. Please find the direct responses to your comments below.

Comments:

Methods:

2.2. Viruses section: More details, including maps, on the generation of MDV reported viruses should be provided. In addition, the RB-1BΔmeq strain should be described.

Response: Thank you for this comment. We included more information on RB-1B, RB-1BΔmeq and CVI988 in paragraph 2.2 and Table S1 as well as a genome map in Figure S1.

Results and discussion:

Page 6, last paragraph. The authors stated that infection rates in NK cells were higher than in primary B and T cells, in primary endothelial cells, and in primary macrophages. However, these data were obtained from studies published in the literature. The authors should perform comparative experiments or, at least, describe more in detail with quantitative data the results reported in the literature.

Response: Thanks for picking this up. We modified the paragraph accordingly and now provide the details and the quantitative data reported in the literature as suggested by the reviewer (paragraph 3.2).

With this, we hope that our manuscript will be favorably reviewed and acceptable for publication in Microorganisms.

Reviewer 2 Report

The authors described that Marek’s disease virus can efficiently infect NK cells and enhances NK cell degranulation and increased IFNγ production in vitro. the study is very interesting but the manuscript is not well-written. There are lots of misspelling and the format is very poor. 

in Figure 1, the authors described that MDV can infect Nk cells, is it productive replication? please show the plaque representative data for Figure 1A. Also, the primers used for Figure 1B should be included in the manuscript. Figure 1 also shown RB-1BΔmeq results, this description of these results should be included in result.

the manuscript must be carefully edited again for instance font size, IFNg should be consistent in the current manuscript.

Author Response

Dear reviewer 2, we would like to thank you for evaluating our manuscript. We have addressed all of your concerns and modified the manuscript accordingly. Please find the direct responses to your comments below.

The authors described that Marek’s disease virus can efficiently infect NK cells and enhances NK cell degranulation and increased IFNγ production in vitro. The study is very interesting but the manuscript is not well-written. There are lots of misspelling and the format is very poor.

Response: Thank you for pointing this out. As suggested by the reviewer, we carefully revised and improved our manuscript. The formatting issue was caused by the conversion to PDF by the journal and was not present in the word file that we uploaded.

in Figure 1, the authors described that MDV can infect NK cells, is it productive replication? Please show the plaque representative data for Figure 1A. Also, the primers used for Figure 1B should be included in the manuscript. Figure 1 also shown RB-1BΔmeq results, this description of these results should be included in result.

Response: As suggested by the reviewer, we included representative plaque images (Fig. S2) and expanded the information on the plaque size assays and growth kinetic experiments in the M&M section (paragraph 2.3). In addition, we included the primer and probe sequences in Table S1. Beyond that, the results of all three reporter viruses are presented in paragraph 3.1 as requested by the reviewer. Our observation that the infection levels in the NK cell population increases over time (Fig. 1C and D) indicates that they are indeed productively infected and pass on the infection. We addressed all these points in the manuscript.

The manuscript must be carefully edited again for instance font size, IFNg should be consistent in the current manuscript.

Response: We edited the manuscript to ensure style consistency and correct grammar, syntax, punctuation and spelling.

With this, we hope that our manuscript will be favorably reviewed and acceptable for publication in Microorganisms.

Round 2

Reviewer 2 Report

the authors made significant changes to the revised manuscript. I have additional minor points, please take into consideration to improve the manuscript.

in M&M, please add company information for fetal bovine serum and chicken serum. 3.1 reporter viruses replicate tocomparable levels. there is space between"to comparable" any statistics test on Figure1C and Figure 1D?  If the authors get NK cells as described in reference 21, the staining should be included CD3. I am just wondering the Figure 2 some parts should show CD3 instead of FSC-A. then it will make sense when looking at Figure 3 showing CD3- cells. In addition, the FSC-A and SSC-A gating for Figures 2D and 2H are a little strange for me. maybe CD45 will help the gating? For 2F, G, J, and K, the large population of cells hide, it might be because the author uses a log scale for GFP gating, can you change to exponential scale as Y-axis.

Author Response

Dear reviewer 2, we would like to thank you for evaluating our manuscript again. We have addressed all of your concerns and modified the manuscript accordingly. Please find the direct responses to your comments below.

The authors made significant changes to the revised manuscript. I have additional minor points, please take into consideration to improve the manuscript.

In M&M, please add company information for fetal bovine serum and chicken serum.

Response: The information was added as suggested.

3.1 reporter viruses replicate tocomparable levels. There is space between "to comparable" any statistics test on Figure 1C and Figure 1D? 

Response: We corrected the sentence as suggested by the reviewer. In Figure 1C and D we use two virus strains (RB-1B and CVI988) that have mild differences in their replication properties. Therefore we don’t feel comfortable comparing them statistically. 

If the authors get NK cells as described in reference 21, the staining should be included CD3. I am just wondering the Figure 2 some parts should show CD3 instead of FSC-A. Then it will make sense when looking at Figure 3 showing CD3- cells. In addition, the FSC-A and SSC-A gating for Figures 2D and 2H are a little strange for me. Maybe CD45 will help the gating? For 2F, G, J, and K, the large population of cells hide, it might be because the author uses a log scale for GFP gating, can you change to exponential scale as Y-axis.

Response: Thanks for this comment! However, in this case we did not include a CD3 staining since we know that CD3 is not expressed on these NK cells isolated from the embryonic spleen. (Jansen et al., 2010). Unfortunately, this point was not clear in our original manuscript. We corrected the labelling of the Y-axis in figure 3. We now state CD107+ NK cells (A, C) and IFNγ+ NK cells with an explanation in the legend that the NK cells were selected based on PKH26+ cells. We have changed the revised manuscript accordingly.

Since CD45 could not be included in the panel we used in the experiments were NK activation was determined (due to limitations in the available channels in our flowcytometer) we were only able to select NK cells based on positive PKH26 labelling and scatter. Based on these parameters, we indeed show a different lymphocyte gate in figure 2D versus H. This is because the expression of CD107 or IFNγ was analyzed in two different samples, since both antibodies are APC-conjugated. Surface expression of CD107 was directly analyzed after the incubation. When plotting CD107 against FSC-A we always observed that only the population of lymphocytes low in FSC-A expressed CD107 (Figure C), suggesting that there is a very small window between degranulation and cell death. This population of FSC-A low cells was selected in figure 2D.

To determine IFNγ production, cells were first fixed and permeabilized, after which an intra cellular staining of IFNγ was performed. Fixation and permeabilization affects the FSC-A/SSC-A profile, which resulted in one lymphocyte population based on FSC-A expression. When IFNγ was plotted against FSC-A, we observed IFNγ in this population. We have included this figure to in Figure 2 and explained the gating in more detail in the figure legend.

For Figure 2 F, G, J and K we have changed the axis from log scale to an exponential scale, which indeed improves the figure.

With this, we hope that our manuscript will be favorably reviewed and acceptable for publication in Microorganisms.